# Parkinson’s Disease Causative Mutation in Vps35 Disturbs Tetherin Trafficking to Cell Surfaces and Facilitates Virus Spread

**DOI:** 10.3390/cells10040746

**Published:** 2021-03-28

**Authors:** Yingzhuo Ding, Yan Li, Gaurav Chhetri, Xiaoxin Peng, Jing Wu, Zejian Wang, Bo Zhao, Wenjuan Zhao, Xueyi Li

**Affiliations:** 1School of Pharmacy, Shanghai Jiao Tong University, Shanghai 200240, China; yingzhuoD@sjtu.edu.cn (Y.D.); 018170210001@sjtu.edu.cn (Y.L.); gvchhetri@sjtu.edu.cn (G.C.); xxpeng@sjtu.edu.cn (X.P.); wjszmwzswy@sjtu.edu.cn (J.W.); wangzejian@sjtu.edu.cn (Z.W.); bozhao@sjtu.edu.cn (B.Z.); zhaowj@sjtu.edu.cn (W.Z.); 2Department of Neurology, Massachusetts General Hospital and Harvard Medical School, Charlestown, MA 02129, USA

**Keywords:** Parkinson’s disease, Vps35, tetherin, herpes simplex virus

## Abstract

Parkinson’s disease (PD) is the most common neurodegenerative movement disorder, characterized by progressive loss of dopaminergic neurons in the substantia nigra, intraneuronal deposition of misfolded proteins known as Lewy bodies, and chronic neuroinflammation. PD can arise from monogenic mutations, but in most cases, the etiology is unclear. Viral infection is gaining increasing attentions as a trigger of PD. In this study, we investigated whether the PD-causative 620 aspartate (D) to asparagine (N) mutation in the vacuolar protein sorting 35 ortholog (Vps35) precipitated herpes simplex virus (HSV) infection. We observed that ectopic expression of Vps35 significantly reduced the proliferation and release of HSV-1 virions; the D620N mutation rendered Vps35 a partial loss of such inhibitory effects. Tetherin is a host cell protein capable of restricting the spread of encapsulated viruses including HSV-1 and SARS-Cov-2, both of which are implicated in the development of parkinsonism. Compared with cells overexpressing wildtype Vps35, cells expressing mutant Vps35 with D620N had less Tetherin on cell surfaces. Real-time and static cell imaging revealed that Tetherin recycled through Vps35-positive endosomes. Expression of Vps35 with D620N reduced endosomal dynamics and frequency of motile Tetherin-containing vesicles, a sign of defective production of recycling carriers. Our study suggests that the D620N mutation in Vps35 hinders Tetherin trafficking to cell surfaces and facilitates virus spread.

## 1. Introduction

Parkinson’s disease (PD), the second most-common, late-onset neurodegenerative disorder after Alzheimer’s disease (AD), is characterized by progressive loss of dopaminergic neurons primarily in the substantia nigra pars compacta, intraneuronal deposition of misfolded protein aggregates known as Lewy bodies, and chronic neuro-inflammation [1]. The prevalence of PD steadily increases at a rate surpassing that of other neurodegenerative disorders including AD, thus leading to warnings of a Parkinson’s pandemic [2,3]. The etiology of PD is largely unknown but considered to involve interplay of aging, genetic and environmental factors [1,4]. A small proportion of PD cases are inherited and arise from mutations in single genes [1,4], among which the vacuolar protein sorting 35 ortholog (Vps35) is linked to a rare, late-onset autosomal-dominant form of PD [5,6]. Although additional Vps35 variants are also segregated with PD, only the aspartate (D) to asparagine (N) mutation at residue 620 is found to be pathogenic [7,8]. Furthermore, Vps35 loss-of-function is involved in the pathogenesis of AD [9,10,11]. Exactly how functional perturbation of Vps35 promotes the pathogenesis of PD as well as AD is still under investigation.

Vps35 is a central component of the retromer complex, which mediates the retrieval of membrane-associated proteins from endosomes to trans-Golgi networks and to the cell surface [2,3,12]. The retromer complex is a heteropentamer composed of a trimeric core to recognize cargo proteins and a sorting nexin heterodimer that deforms endosomal membranes to generate recycling tubules and/or vesicles [13,14,15]. Vps35 acts as a scaffold for Vps26 binding at the N-terminal domain and Vps29 binding at the C-terminal domain to form the cargo-recognition core of the retromer [15,16]. The D620N mutation in Vps35 does not affect the formation of the Vps26/Vps35/Vps29 heterodimer nor alter cargo recognition [17]. Ectopically expressed as well as endogenous Vps35 with the D620N mutation induces the enlargement of retromer-positive endosomes and impairs the formation of retromer transport carriers and retrieval of dopamine D1 receptors, dopamine transporters, and many other proteins, including those involved in the degradation of α-synuclein [17,18,19,20,21]. The presence of D620N in Vps35 also impairs autophagy and causes accumulation of PD-variant α-synuclein A53T [22]. Upregulation of Vps35 reduces the intraneuronal accumulation of α-synuclein, whereas expression of PD-linked Vps35 P316S or knockdown Vps35 increases α-synuclein accumulation [23]. These observations suggest that Vps35 D620N may impair the clearance of α-synuclein, thereby leading to deposition of α-synuclein. However, accumulation of α-synuclein was not apparent in Vps35 D620N knock-in mice [24]. Collectively, the disease-causing effects of Vps35 D620N may involve other mechanism(s) or facilitator(s).

The infectious etiology of PD is regaining considerable attention, particularly with the report of a case developing parkinsonian traits after infection with the current pandemic’s SARS-CoV-2, or COVID-19 [25,26,27]. The hypothesis of an infectious origin of PD was put forward originally based on observations of parkinsonism in 7 individuals who developed encephalopathies following the 1918 influenza pandemic. Later on, “dual-hit” and “three-phase” hypotheses regarding the contribution of infectious pathogens to the development of PD were proposed [28,29]. Herpes simplex virus type 1 (HSV-1) is an encapsulated, double-stranded DNA virus that infects over half the global population and typically resides in peripheral neurons in a non-replicative latent state [28]. The ability of HSV-1 to maintain a latent state allows for lifelong infection. HSV-1 reactivation can be asymptomatic and occasionally causes herpetic encephalitis [29]. An association between HSV-1 infection/reactivation and PD has been demonstrated in epidemiological and case-control studies [30]. Exactly how viral infection contributes to the pathogenesis of PD remains to be defined. Intranasal inoculation of viruses in mice has been shown to cause deposition of aggregated α-synuclein and degeneration of dopaminergic neurons [31,32,33]. Viral infection, either systemic or locally reactivated, may also escalate neuroinflammation by inducing a significant “cytokine storm” [34].

Tetherin is an interferon-inducible host protein doubly linked to membranes through its N-terminal transmembrane domain and C-terminal glycosyl–phosphatidyl–inositol moiety [35]. This unusual topology of Tetherin enables it to serve as a tether with one end linked to the plasma membrane of the host cell and the other end inserted into the envelope of the budding virion, thus mobilizing nascent virions in close proximity to the cell surface of the host cell and preventing the spread of nascent virions to neighbor cells [36]. Viruses produce factors driving Tetherin internalization to counteract the virus-restricting activity of Tetherin [37]. In virus-free cells, Tetherin undergoes clathrin-dependent endocytosis and travels to trans-Golgi networks in a clathrin-dependent manner [38]. Endocytosed Tetherin is also found at structures containing cation-independent mannose-6-phosphate receptor [38], a typical cargo of retromer [39]. These observations implicate that Tetherin may recycle through the retromer-dependent trafficking pathway. Tetherin can be expressed in primary neurons as well as in the brains of mice upon exposure to interferon-α and/or infection with a neurotropic virus. However, whether Tetherin travels through the retromer-labeled pathway and has a link to PD as well as other neurological disorders has not been examined to date.

In this study, we investigated whether the PD-causal D620N mutation in Vps35 affected Tetherin trafficking and/or HSV-1 spread. We found that ectopic expression of wildtype Vps35 markedly decreased the titer of HSV-1 virions inside as well as in culture media of HSV-1-infected cells. Vps35 with the D620N mutation exhibited similar inhibitory effects on HSV-1 proliferation and release but to a much lesser extent than wildtype Vps35. Ectopic expression of Vps35 with or without the D620N mutation did not alter the overall level of Tetherin, but mutant Vps35 caused reduction of Tetherin on the cell surface when compared with wildtype Vps35. Time-lapse live cell imaging revealed that D620N in Vps35 triggered a decline of endosomal dynamics. Our study suggests that the Parkinson’s disease-causative D620N mutation in Vps35 impedes Tetherin going back onto cell surfaces and facilitates the spread of HSV-1.

## 2. Materials and Methods

### 2.1. Plasmids

The cDNA-encoding human Tetherin were amplified by reverse transcriptase polymerase chain reaction (RT-PCR) with 5′ gtctgctagccaccatggcatctacttcgtatgac and 5′ tgtgagtctcgagatctcactgcagcagagcgct and cloned into Nhe I and Xho I sites of pcDNA_3_. To generate the construct expressing-Tetherin–eGFPc, the N-terminal and C-terminal parts of Tetherin were amplified from pcDNA_3_-Tetherin by PCR with primers (Nf: 5′ gtctgctagccaccatggcatctacttcgtatgac and Nr: 5′ gtcggtacctttcttttgtccttgggccttctc; Cf: 5′ tgtgagtcgaattcgtggaggagcttgagggagag and Cr: 5′ tgtgagtctcgagatctcactgcagcagagcgct). The N-terminal Tetherin PCR fragment was digested with Nhe I and Kpn I, whereas the C-terminal fragment was treated with EcoR I and Xho I. The enhanced green fluorescent protein (eGFP) open-reading frame (ORF) amplified by PCR from pEGFP–N1 was treated with Kpn I and EcoR I, ligated with both Tetherin DNA fragments, and cloned into the Nhe I and Xho I sites of pcDNA_3_. To generate the construct-expressing Arf6 with dsRed fused to the C-terminus, the cDNA for human Arf6 was amplified by RT-PCR with 5′ tttggtaccatggggaaggtgctatcca and 5′ cgaccggtgcagatttgtagttagaggtta, digested with Kpn I and Age I, and cloned along with the dsRed ORF treated with Age I and Xho I into the Kpn I and Xho I sites of pcDNA_3_. To construct the plasmid-expressing Rab4, Rab5, Rab11, and Vps35 with dsRed fused to the N-terminus, the ORF of dsRed was amplified by PCR from pdsRed-C1; digested with Nhe I and Bam HI; ligated with the Bam HI and Xho I-digested ORF for each of Rab4, Rab5, and Rab11; and cloned into Nhe I and Xho I sites of pcDNA_3_, whereas the cDNA for human Vps35 was amplified with 5′ atatggatccatgcctacaacacagcagtc and 5′ ggggctcgagttaaaggatgagaccttcataaat, digested with Bam HI and Xho I, and cloned into the Bgl II and Xho I sites of pdsRed-C1. The dsRed in pcDNA_3_–dsRed–Rab11 was replaced with mCherry to generate pcDNA_3_–mCherry–Rab11. Vps35 was also cloned along with a pair of oligoes encoding a FLAG (DYKDDDDK) tag into pcDNA_3_. The D620N mutation was generated using PCR with pcDNA_3_–FLAG–Vps35 and/or pdsRed–Vps35 as templates using primers (5′ gaagatgaaatcagcaattccaaagcaaagcacagc and 5′ gctgtgctttggaattgctgatttcatcttc). All constructs were confirmed by DNA sequencing before being used for studies.

### 2.2. Cell Culture and Transfection

HeLa, HEK293T, and Vero cells were maintained in dulbecco’s modified eagle medium (DMEM) supplemented with 10% fetal bovine serum, L-glutamine, and penicillin/streptomycin at 37 °C in a humidified cell culture incubator (ThermoFisher Scientific China, Shanghai, China). DMEM medium, fetal bovine serum, and supplements were purchased from ThermoFisher Scientific China. Transfection was conducted with Lipofectamine™ 2000 (ThermoFisher Scientific China) according to the supplier’s instructions. The day before transfection, cells were plated in a 6-well plate with (microscopy) or without (viral infection) coverslips or in 60-mm dishes at 1 × 10^5^/mL. Cells for time-lapse imaging were seeded in glass bottom cell culture dishes (NEST Biotechnology, Wuxi, China). Before preparing liposome–DNA mixtures, cells were changed into Opti-MEM medium (ThermoFisher) and cultured at 37 °C for 2 h in a cell culture incubator. Liposome–plasmid mixtures were incubated for 25 min, with gentle mixing every 5 min, and added to each well or plates. After gently rocking plates back and forth several times, cells were cultured in Opti-MEM media for 6 h and then changed into complete DMEM media. Cells were cultured until being processed for studies. Usually, transfection efficiency was 60% to 80%.

For biochemical examination of the distribution and glycosylation of Tetherin-eGFPc, we utilized 2 μg of pcDNA_3_–Tetherin–eGFPc and 2 μL of Lipofectamine™ 2000 for each transfection of cells in a 60-mm dish. We utilized 2 μL of Lipofectamine™ 2000 per 1 μg of plasmids for cells subjected to microscopic imaging. To determine whether ectopic expression of Vps35 or Vps35D620N affected Tetherin on cell surfaces, 0.5 μg of pdsRed–Vps35 or pdsRed–Vps35D620N were used for each transfection. For live cell imaging, 0.5 μg of pcDNA_3_–Tetherin–eGFPc along with 0.5 μg of pdsRed–Vps35 or pdsRed–Vps35D620N were used for each transfection. For examining the localization of Tetherin–GFPc at endosomal compartments, 0.5 μg of pcDNA_3_–Tetherin–eGFPc for each transfection were mixed with 0.5 μg of pdsRed–Vps35, pcDNA_3_–dsRed–Rab4, pcDNA_3_–dsRed–Rab5, pcDNA_3_–mCherry–Rab11, or pcDNA3–dsRed–Arf6. Cells used for microscopy were treated with 50 μg mL^−1^ of β-cycloheximide for 5 h.

### 2.3. Virus Titer Determination

HSV-1 was propagated in Vero cells and stored in aliquots at −80 °C until use for studies. To facilitate the measurement of the absolute copy number of HSV-1 virions, a 526 bp fragment was amplified from *UL30* of HSV1 with primers (5′ gcgtttatcaaccgcacctc and 5′ aatagtccgtgttcagggcg) and cloned into pdsRed (annotated as pdsRed-UL30). Titers of HSV-1 stocks and virions in conditioned media of infected cells were expressed as copy numbers of *UL30*, which were determined with reference to a standard curve derived using pdsRed–UL30 as quantitative PCR (qPCR) templates. We determined titers of HSV-1 inside cells by semi-quantitative PCR amplification of *UL30* using *GAPDH* as an internal control. qPCR was conducted using FastStart Universal SYBR Green Master (Roche, Basel, Switzerland) in an ABI StepOnePlus Real-Time PCR System (ThermoFisher). We exploited the comparative C_T_ (ΔΔC_T_) method to determine the change in copies of *UL30* between treatment conditions, with GAPDH as an internal standard. Copy numbers were normalized with the volume of conditioned media or with the amount of genomic DNAs used for qPCR. Specific qPCR primers were 5′ catcaccgacccggagagggac and 5′ gggccaggcgcttgttggtgta for *UL30* and 5′ ctgggctacactgagcacc and 5′aagtggtcgttgagggcaatg for *GAPDH*.

### 2.4. HSV-1 Entry, Replication, and Release Assays

HeLa or HEK293T cells were infected with an aliquot of HSV-1 stock at an indicated multiplicity of infection (MOI). To determine the effects of Tetherin, Vps35, Vps35D620N, or their combination on HSV-1 entry, we harvested cells 2 h after exposure to HSV-1 and prepared genomic DNAs for semi-qPCR quantification of *UL30* with GAPDH as an internal control. HSV-1 replication was determined by semi-qPCR quantification of *UL30* in genomic DNAs prepared from cells harvested 36 h after exposure to HSV-1. We assayed HSV-1 release by measuring the absolute copy number of *UL30* in conditioned media and by semi-qPCR quantification of *UL30* in genomic DNAs prepared from cells exposed to conditioned media for 2 h.

### 2.5. Subcellular Fractionation

Subcellular fractionation was performed as described previously. In brief, cells were cultured in the presence or absence of 50 μg mL^–1^ of β-cycloheximide for 5 h and then scraped into media with the use of a disposable cell scraper. Cells were spun down at 4 °C and 500× *g* for 5 min and washed sequentially with cold PBS and ice-cold homogenization buffer (25 mM Tris/Cl pH7.5 and 130 mM KCl). Cell pellets were resuspended in homogenization buffer containing protease inhibitors and homogenized by passing through a 25-gauge needle 20 times on ice. Lysates were cleared by a centrifugation at 4 °C and 1000× *g* for 5 min. Post-nuclear supernatants were overlaid on a discontinuous Nycodenz gradient pre-made in an SW41 tube (Beckman Coulter, Indianapolis, IN, USA) as 0.66 mL of 40% and 5 mL each of 25% and 5% Nycodenz (Axis-Shield/Alere Technologies AS, Oslo, Norway). After gradients were centrifuged at 4 °C and 30,000 rpm for 1 h, 13 fractions at 1 mL each, were collected from top to bottom of each gradient. Equal volumes of each fraction were used for precipitating proteins with 3 volumes of methanol/chloroform (2:1, *v*/*v*). After being dried at 37 °C for 1 h, precipitated proteins in each fraction were used for Western blot.

### 2.6. Western Blot

Western blot analysis was performed according to standard procedures. Primary antibodies used for Western blot analysis included rabbit anti-Vps35 (1:5000, Abcam, ab226180), rabbit anti-BST2 (1:1000, Abcam, Cambridge, MA, USA, ab134061), mouse anti-FLAG (1:5000, CMCTAG, Inc., Milwaukee, WI, USA, AT0022), and mouse anti-GAPDH (1:5000; ProteinTech, Rosemont, IL, USA, 60004-1-Ig). All secondary antibodies used for Western blot analysis were diluted at 1:5000. Peroxidase AffiniPure goat anti-mouse (115035003) and anti-rabbit (111035003) IgG(H+L) were purchased from Jackson ImmunoResearch (West Grove, USA). Blots were developed using enhanced Pierce^TM^ enhanced chemiluminescence (ThermoFisher) and imaged with the ChemiDocTM MP Imaging System (Bio-Rad, Hercules, CA, USA). Densitometry was conducted using NIH ImageJ/Fiji software 1.48v.

### 2.7. Immunofluorescence Microscopy and Time-Lapse Live Cell Imaging

Transfected HeLa cells on glass coverslips were fixed in 4% paraformaldehyde or in pre-cooled methanol. To examine levels of Tetherin on cell surfaces, we omitted detergent in all buffers used for immunolabeling and wash steps. HeLa cells on glass coverslips were fixed in 4% paraformaldehyde at room temperature for 15 min, washed in PBS, blocked in PBS containing 2% bovine serum albumin, and incubated with antibodies against BST2 (1:100, Abcam, ab243229) at 4 °C overnight. After washes in PBS, cells were incubated with Alexa Fluor^®^ 594-conjugated AffiniPure goat anti-rabbit IgG (H+L) (1:1000, Jackson Laboratory, 111585003) and Hoechst 33258 (1:2000, Invitrogen/ThermoFisher, H3569). Digital images of each channel were collected separately using the Zeiss LSM710 or the Leica TCS SP8 confocal imaging system with the same imaging settings including laser strength, pinhole, signal gain, resolution, and scan times.

For live cell imaging, transfected HeLa cells in glass bottom cell culture dishes were changed into phenol red-free media containing 50 μg mL^−1^ of β-cycloheximide. Time-lapse imaging was performed using a Zeiss LSM 900 confocal laser scanning microscope equipped with an Airyscan with live cell capabilities and fitted with a fast-AS module (Carl Zeiss AG, Oberkochen, Germany). The NA oil-immersion alpha Plan-Apochromat 63x/1.40 Oil CorrM27 objective (Zeiss) was used with Immersol 518 F immersion media (Carl Zeiss AG) and Zen Blue software 2.1. We used 488 nm excitation (green) and 561 nm excitation (red) for imaging the Tetherin-eGFPc and dsRed-Vps35 or dsRed-Vps35D620N, respectively. Images were acquired at intervals of 10 s at a pixel resolution of 0.10 μm (fast-AS mode) in XY and at intervals of 0.2 μm in Z, step-size using the piezo drive. The microscope was equipped with an environmental chamber that was maintained at 37 °C with humidified 5% CO_2_ gas during the entire imaging period of 5 min.

### 2.8. Image Analysis

Images were analyzed using NIH ImageJ/Fiji software by at least two examiners who were blinded to experimental conditions. The JACoP plugin was applied to determine the Pearson’s correlation coefficient as a measure of the frequency of protein co-localization in cells. Images used for co-localization analysis were obtained from at least 3 independent transfections or experiments. To quantify levels of Tetherin on cell surfaces, images were converted to 8-bit black-and-white images. After background signals were removed, the edge of each cell in a particular image was tracked using the freehand tool to generate a contour. Signal intensities within the contour were measured. Data were represented as Mean ± SD of signal intensities per cell.

We determined the dynamics of structures containing Tetherin–eGFPc and dsRed–Vps35 in two ways. In one, we counted the number of motile structures, which were defined as those present in a previous image but absent in following one(s), and vice versa, and/or those moving from one place to another. In the other, we tracked the change of their size and or morphology. The size distributions of endosomal structures labeled with Tetherin–eGFPc and dsRed–Vps35 were determined as previously described [40]. In brief, movies for each of the imaged cells were segmented. A series of 9 consecutive frames from segmented movies of cells expressing Vps35 or Vps35D620N were chosen for measuring the size of endosomes. The brightness and contrast of the images were optimized to visualize individual endosomes as well as small vesicles. The imaged cell was outlined using the freehand tool of ImageJ/Fiji software, and the surrounding areas were removed. We applied the “Analyze Particles” function to obtain the cross-section area scale of each structure within the outline (imaged cell). The value of each measured particle was converted into nm^2^ based on the original size and resolution of the image. We grouped them based on their size in nm^2^ and calculated their percentage for graphing and comparison.

### 2.9. Statistical Analysis

We conducted statistical analyses using GraphPad Prism 7 software. All data were represented as Mean ± SD and graphed in GraphPad Prism 7. An unpaired, two-tailed Student’s *t*-test was used for comparison between two conditions, whereas one-way ANOVA followed by Turkey’s test was conducted to determine statistical significance among three different treatment conditions. *p* < 0.05 was statistically significant.

## 3. Results

### 3.1. The D620N Mutation Dampens Vps35 in Constraining HSV-1 Propagation

Previous studies have demonstrated that Vps35 regulates the delivery of HIV-1 envelope glycoprotein for incorporation into nascent virions, the entry of human papillomavirus into host cells, and the replication of hepatitis virus C in host cells [41,42,43,44]. To examine whether Vps35 also modulates HSV-1 infection, we transfected HeLa cells with plasmids expressing DsRed-fused Vps35 or Vps35D620N and then exposed them to HSV-1 (Figure 1A). Ectopically expressed dsRed-Vps35 was co-precipitated with endogenous Vps26 (Appendix A), indicating the functionality of dsRed-Vps35 in the retromer complex. Titers of HSV-1 inside as well as in cell culture supernatants (conditioned media) of infected cells at different times post-viral infection were determined by quantitative PCR (qPCR) amplification of a fragment in the *UL30* gene of HSV-1. We found that cells transfected with Vps35-expressing plasmids and cells transfected with empty plasmids contained comparable copies of *UL30* when cells were exposed to HSV-1 for 2 h (Figure 1B), suggesting that ectopic expression of Vps35 did not affect HSV-1 entry into host cells. However, 36 h after HSV-1 infection, cells transfected with Vps35-expressing plasmids had significantly less copies of *UL30* than cells transfected with empty plasmids (Figure 1C), supporting the idea that Vps35 suffocates HSV-1 replication. To examine whether overexpression of Vps35 affected the release of HSV-1 virions, we harvested conditioned media 36 h after viral infection and found that copies of *UL30* in conditioned media of cells ectopically expressing Vps35 were significantly reduced relative to those of cells transfected with empty vectors (Figure 1D). To demonstrate that the *UL30* gene in conditioned media originated from infectious virions instead of naked DNAs, we incubated HeLa cells with conditioned media for 2 h. After extensive washes to remove trace amounts of conditioned media, cells were harvested for preparing genomic DNAs. Under these conditions, we detected *UL30* at expected copies in DNA samples of cells exposed to conditioned media (Figure 1E). Collectively, these data showed the ability of Vps35 to dampen HSV-1 propagation.

The D620N mutation in Vps35 is unambiguously identified as a cause of a late-onset, autosomal-dominant form of PD. While the mechanisms involved are not completely understood, the D620N mutation has been shown to confer toxic functions as well as cause loss of normal functions [7]. HSV-1 infection/reactivation is gaining increasing attention as a risk factor for neurodegenerative diseases including PD [45]. To determine whether the D620N mutation in Vps35 potentiates HSV-1 propagation, we measured copies of *UL30* inside and in conditioned media of cells transfected with plasmids expressing Vps35D620N after exposure to HSV-1. Our results showed that similar to ectopic expression of wildtype Vps35, expression of Vps35D620N had little effect on HSV-1 entry, but it significantly reduced the proliferation and release of HSV-1 virions (Figure 1B–E). We noticed that copies of *UL30* inside and in conditioned media of cells expressing Vps35D620N were significantly increased when compared with those of cells ectopically expressing wildtype Vps35 (Figure 1C–E). We asked whether the Vps35D620N-induced increase in copies of *UL30* was simply a consequence of poor expression of Vp35D620N and/or resulted from compromised expression of endogenous Vps35. Our data revealed that exogenous Vps35 and Vps35D620N were expressed at comparable levels, and that their presence did not alter the expression of endogenous Vps35 (Figure 2A,B). Taken together, these data suggest that the D620N mutation in Vps35 causes a partial loss of function for HSV-1 propagation.

### 3.2. Tetherin Mediates the Inhibitory Effect of Vps35 on HSV-1 Propagation

Tetherin is a host protein that prevents the spread of a wide range of enveloped viruses including HSV-1. We speculated that Tetherin might mediate the regulatory effect of Vps35 on HSV-1 propagation. We first conducted Western blot analysis to determine whether ectopic expression of Vps35 with or without D620N interfered with the expression of endogenous Tetherin in HeLa cells and found that the overall level of Tetherin was comparable between cells ectopically expressing Vps35 and cells expressing Vps35D620N (Figure 2A,B). We then compared levels of Tetherin on the cell surface among cells expressing Vps35, cells expressing Vps35D620N, and cells expressing dsRed only. We employed unpermeabilized cells for immunolabeling Tetherin to detect only Tetherin molecules on and close to cell surfaces. Immunofluorescence microscopy followed by densitometry analysis revealed that signal intensities of Tetherin in cells expressing Vps35D620N significantly declined relative to those in cells ectopically expressing Vps35 (Figure 2C,D). These data suggest that Vps35D620N redistributes Tetherin away from plasma membranes, thereby making Tetherin unable to restrict HSV-1 virions.

To further demonstrate that Tetherin contributes to the regulatory effect of Vps35 on HSV-1 propagation, we took advantage of HEK293T cells, which do not express Tetherin (Figure 3A). We made a side-by-side comparison of HSV-1 replication and release of HEK293T cells with those in HeLa cells, which constitutively express Tetherin (Figure 2A). HeLa and HEK293T cells were infected with the same aliquot of HSV-1 stock at the same multiplicity of infection (MOI). We harvested HeLa and HEK293T cells and their conditioned media 36 h after infection with HSV-1 for analyses. Ectopic expression of Tetherin or Vps35 did not affect the expression of endogenous Vps35 (Figure 2B and Figure 3B). Under these conditions, we found that HSV-1 infected HEK293T cells contained and released significantly more virions than HSV-1 infected HeLa cells (Appendix A). Ectopic expression of Tetherin in HEK293T cells vastly reduced the copies of *UL30* inside and in conditioned media of HSV-1 infected HEK293T cells (Figure 3C,D). Co-expression of Vps35 or Vps35D620N with Tetherin led to a further decrease in *UL30* copies (Figure 3C,D), thus enhancing the activity of Tetherin in restricting HSV-1 propagation. We noticed that there were more copies of *UL30* in conditioned media of HEK293T cells co-transfected with plasmids expressing Tetherin and Vps35D620N than in conditioned media of HEK293T cells co-expressing exogenous Tetherin and wildtype Vps35 (Figure 3C,D). The *UL30* gene in the conditioned media of HEK293T cells originated mainly from bona fide infectious HSV-1 virions rather than from naked HSV-1 DNAs (Figure 3E). Collectively, these data suggest that the D620N mutation causes Vps35 to partially lose its inhibitory effect on HSV-1 spread by shifting Tetherin away from cell surfaces.

### 3.3. Tetherin Traffics Together with Vps35

As Vps35 plays a critical role in recycling membrane-associated proteins from endosomes [2] and Vps35D620N reduces Tetherin on cell surfaces (Figure 2A–D), we reasoned that the D620N mutation interfered with Vps35 in regulating Tetherin recycling. To facilitate the investigation of Tetherin trafficking in cells, we constructed a reporter by inserting an enhanced green fluorescent protein (eGFP) moiety into Tetherin, annotated as Tetherin-eGFPc (Figure 4A). The insertion site was chosen based on the difference between Tetherin and its hamster homolog GREG (*G*olgi-*RE*sident *G*PI-anchored protein), which contains three tandem EQEAQIK repeats [46]. We verified that Tetherin-eGFPc mimicked endogenous Tetherin in two ways (Figure 4B,C). In the first, eGFP insertion did not affect glycosylation of Tetherin. In agreement with previous findings [35,46], Tetherin migrated as smears on SDS-PAGE gels (Figure 4B,C). Upon treatment with PNGase-F, which removes glycans attached to asparagine residues, Tetherin smears disappeared, and a single protein band was detected by Tetherin-specific antibodies (Figure 4B). Under these conditions, Tetherin–eGFPc was also detected as smears that disappeared upon treatment with PNGase-F (Figure 4B). In the second, Tetherin–eGFPc had a distribution pattern overlapping well with that of endogenous Tetherin in density gradient ultracentrifugation fractions (Figure 4C).

Having demonstrated that Tetherin–eGFPc mimics endogenous Tetherin, we then conducted time-lapse live cell imaging to determine whether Tetherin trafficked together with Vps35 and or through Vps35 positive endosomes. HeLa cells were co-transfected with plasmids expressing Tetherin–eGFPc and plasmids expressing dsRed–Vps35 or dsRed–Vps35D620N. Prior to subjecting to and throughout the whole duration of time-lapse imaging, cells were exposed to β-cycloheximide to inhibit de novo protein synthesis so as to maximally detect the recycling Tetherin–eGFPc molecules. Under these experimental conditions, we found that Tetherin–eGFPc was co-localized with dsRed–Vps35 with or without D620N on large tubulovesicular structures as well as highly motile small spherical structures or vesicles (Figure 5, Appendix A). Highly motile vesicles containing both Tetherin–eGFPc and dsRed–Vps35 were detectable in both cells expressing wildtype Vps35 and cells expressing Vps35D620N (Figure 5, Appendix A). Quantification of motile vesicles over the whole imaging period revealed that their frequency in cells expressing Vps35D620N was significantly decreased when compared with that in cells expressing wildtype Vps35 (*n* = 3 movies per condition; Mean ± SD, Vps35 vs. Vps35D620N; number of motile structures: 95.7 ± 32.5 vs. 22.7 ± 9.9, *p* < 0.05; total structures: 218.7 ± 88.9 vs. 271 ± 141, *p* = 0.615). In addition, large tubulovesicular structures containing Tetherin–eGFPc and dsRed–Vps35D620N were relatively stationary and less dynamic when compared with those containing Tetherin–eGFPc and dsRed–Vps35 (Figure 5, Appendix A). These data indicate that Tetherin recycles through the retromer-labeled pathway and suggest that the D620N mutation in Vps35 perturbs endosomal dynamics and is likely to impair the production of recycling vesicles.

### 3.4. Tetherin Recycles Mainly through the Vps35 Dependent Trafficking Pathway

Endocytosed proteins can be retrieved through three pathways: Rab4-dependent fast recycling directly from early endosomes, Rab11-dependent slow recycling through perinuclear recycling endosomes, and retromer-dependent retrograde transport from early endosomes to trans-Golgi networks [47]. To further examine the recycling pathways of Tetherin, we transfected HeLa cells with the Tetherin–eGFPc-expressing plasmid together with plasmids expressing each of dsRed-tagged Vps35, Rab4, Rab5, Rab11, and Arf6, which demarcated different endosomal compartments. After treatment with β-cycloheximide for 5 h, cells were processed for fluorescent microscopy. Consistent with findings in time-lapse live cell imaging studies, Tetherin–eGFPc was well co-localized with dsRed–Vps35 and dsRed–Vps35D620N in fixed cells, but Tetherin–GFPc and dsRed–Vps35D620N signals were concentrated at large structures compared with diffusely distributed structures containing both Tetherin–GFPc and dsRed–Vps35 (Figure 6, Appendix A). Arf6 participated in clathrin-dependent and clathrin-independent endocytosis as well as post-endocytic recycling at recycling endosomes [48]. Co-localization of Tetherin–eGFPc with dsRed–Arf6 occurred at small spherical structures near plasma membranes, some of which might have been newly formed endocytic vesicles (Figure 6). We also observed a proportion of Tetherin–eGFPc located at spherical structures positive for dsRed–Rab4 (Figure 6). Co-localization between Tetherin–eGFPc and dsRed–Rab5 was rare, suggesting that endocytosed Tetherin–eGFPc molecules were rapidly sorted out from early endosomes into recycling pathways. We found that signals for dsRed–Rab11 were concentrated at a few large punctate structures (Appendix A); this distribution pattern of dsRed–Rab11 signals markedly differed from a relatively diffuse pattern of endogenous Rab11 [49]. Therefore, we replaced dsRed–Rab11 with mCherry–Rab11, which exhibits a relatively diffuse pattern similar to that of endogenous Rab11 and has been used for tracking Rab11 endosomes [50]. Under these conditions, we found that the frequency of co-localization between mCherry–Rab11 and Tetherin–eGFPc was quite low, though NIH ImageJ based JACoP analysis suggested a significant proportion of Tetherin located at Rab11 positive recycling endosomes (Figure 6). Tetherin-eGFPc was partially co-localized with Golgi marker GM130 (Appendix A). Collectively, these data suggest that at steady state, Tetherin recycles mainly through the retromer-dependent pathway, with a minor proportion going back to cell surfaces via Rab4-dependent fast as well as Rab11-controled slow recycling.

## 4. Discussion

A heterozygous D620N mutation in VPS35, a key subunit of the retromer complex, was linked to a late-onset autosomal dominant form of PD in large kindreds of different ethnic backgrounds [5,6]. Subsequent studies reveal that this mutation is relatively rare, with a calculated prevalence of approximately 0.115% [51]. Individuals with the D620N mutation in Vps35 presents symptoms and pathological signs indistinguishable from those of sporadic PD [5,6]. The D620N mutation in Vps35 is also associated with sporadic PD [52]. These findings make Vps35 a good candidate for studying PD in humans. How Vps35 D620N triggers neurodegeneration is still undetermined, but it is suggested to involve impaired function of retromers and defective retrieval of proteins critical for the function and homeostasis of neurons. Previous studies implicate but do not formally characterize whether the host virus-restricting factor Tetherin traffics in a retromer-dependent manner or pathway [38]. In this study, we report that Tetherin undergoes recycling via the retromer-labeled transport pathway. However, it is not clear whether Tetherin directly binds to Vps35 or the cargo recognition core of the retromer, as only the NH_2_ terminal short tail of Tetherin is located in the cytoplasm and does not contain a known signal, e.g., [F/Y/W]X[L/M/V], for recognition by the retromer [3]. Using real-time and static cell imaging, we found that endocytosed Tetherin was mainly localized at structures containing Vps35.

The PD-linked D620N mutation in Vps35 diminished Tetherin on the cell surface, likely by perturbing the production of cargo-laden recycling vesicles from retromer endosomes. Our live cell imaging revealed that retromer endosomes labeled by Vps35D620N were less dynamic than those labeled by wildtype Vps35 and detected less frequently small, motile, Tetherin-containing vesicles in cells expressing mutant Vps35 with D620N than in cells expressing wildtype Vps35. These observations are consistent with recent findings showing that Vps35D620N impairs the production of transport carriers [21].

Genetic studies have demonstrated that the D620N mutation may not be fully penetrated and can occur in unaffected carriers [5,6,52], implicating that the pathogenicity of D620N in some cases may require the concerted actions of other factor(s) or facilitator(s). In this regard, decreased expression of Tetherin on cell surfaces in the presence of D620N can contribute significantly to the development of parkinsonism. Tetherin restricts the spread of a broad spectrum of enveloped viruses including those implicated in PD, e.g., HSV-1, SARS-Cov-2, and hepatitis virus C [53,54,55]. Reduction of Tetherin on the surface of cells in the brain of D620N positive individuals at risk for PD may cause wide spread of neurotropic viruses, particularly those quiescently persisting in the brain, and accelerate disease development. Consistent with this idea, we found that cells expressing mutant Vps35 with D620N contained and released more copies of HSV-1 virions than cells expressing wildtype Vps35.

Our results suggested that Tetherin made an important contribution to the inhibitory effect of Vps35 on HSV-1 propagation. We found that HEK293T cells, which do not express Tetherin, contained and released more HSV-1 virions than HeLa cells, which constitutively express Tetherin. Introduction of exogenous Tetherin into HEK293T cells dampened the replication and release of HSV-1 virions. Co-expression of Vps35 with Tetherin caused a further decrease in copies of HSV-1 virions inside HEK293T cells and released into culture media from HSV-1 infected HEK293T cells. However, overexpression of Tetherin alone in HEK293T cells did not produce the same effect on HSV-1 propagation as co-expression with Vps35 or with Vps35D620N, suggesting that the inhibitory effect of Vps35 on virus propagation involved other factor(s). Vps35 has been shown to regulate the replication, assembly, and release of virions [41,42,43,44]. The inhibitory effect of Vps35 on HSV-1 propagation might also arise from its regulation on the trafficking of HSV-1 envelope proteins required for the assembly and release of HSV-1 virions and or on the trafficking of other host virus-restricting factor(s). Based on a model summarized by Ahmad and Wilson [56], envelopment of naïve HSV-1 capsids at trans-Golgi networks requires at least the core fusion machinery gB, gD, and gH/gL, which are endocytosed from plasma membranes of host cells and delivered to trans-Golgi networks. While gB is known to contain signals for endocytosis and retrieval to trans-Golgi networks, gD and gH/gL do not contain a recognizable endocytic signal and rely on other viral proteins, e.g., gK/UL20 and gM, for endocytosis and delivery to trans-Golgi networks [56,57]. We analyzed the cytoplasmic sequences and found retrieval signals for recognition by retromer, e.g., FDV (aa336-338) in gK and multiple [F/Y/W]X[L/M/V] motifs in gM. Future studies are needed to characterize whether gK and gM are cargoes of retromers.

Evidence is available supporting that neurodegeneration induced by the D620N mutation involves loss-of-function and or gain-of-function mechanisms [24,58]. Our results showed that while both wildtype and mutant Vps35 were capable of dampening HSV-1 replication and release, the inhibitory effect of mutant Vps35 on HSV-1 replication and release markedly declined relative to that of wildtype Vps35. Hence, the D620N mutation in Vps35 rendered a loss-of-function effect on constraining viral propagation. Although their pathogenicity remains inconclusive, several other Vps35 variants are also segregated with PD [8]. It is possible that the pathogenicity of these variants may commence when recurrent viral infection occurs. Future studies are necessary to examine whether these Vps35 variants affect Tetherin trafficking and/or virus propagation in a way similar to the D620N variant.

As viral infection has been shown to upregulate amyloidogenic α-synuclein species and cause deposition of aggregated α-synuclein [31,32,33], and D620N reduces the inhibitory effect of Vps35 on viral propagation, brain cells bearing Vps35D620N are prone to virus spread, which may plausibly increase the production of phosphorylated α-synuclein and lead to the deposition of α-synuclein aggregates. Contrary to this speculation, aggregation of α-synuclein was not evidenced in the brain of Vps35D620N knock-in mice [24]. It is not clear whether the rare formation of α-synuclein inclusions in the brains of Vps35D620N knock-in mice is a consequence of lacking the challenge of environmental pathogens, as these mice were housed in a pathogen-free environment.

In summary, we suggested that Tetherin recycled via the retromer positive pathway and showed that the PD-causative mutation in Vps35 impairs Tetherin recycling back onto cell surfaces, and as a return, facilitates virus spread. Our study suggests that recurrent HSV-1 infection in the brain synergizes with Vps35 D620N to magnify the damage of brain cells. However, our studies have limitations in understanding the pathogenesis of PD, as we utilized non-neuronal cell lines to investigate effects of transiently expressed Vps35 and its pathogenic D620N variant on Tetherin trafficking and HSV-1 propagation. In addition, HSV-1 infection as a risk factor is relatively more relevant to AD than to PD. Nevertheless, our study provides an important clue for future studies investigating the potential involvement of Tetherin in the pathogenesis of PD as well as AD.

## Figures and Tables

**Figure 1 cells-10-00746-f001:**
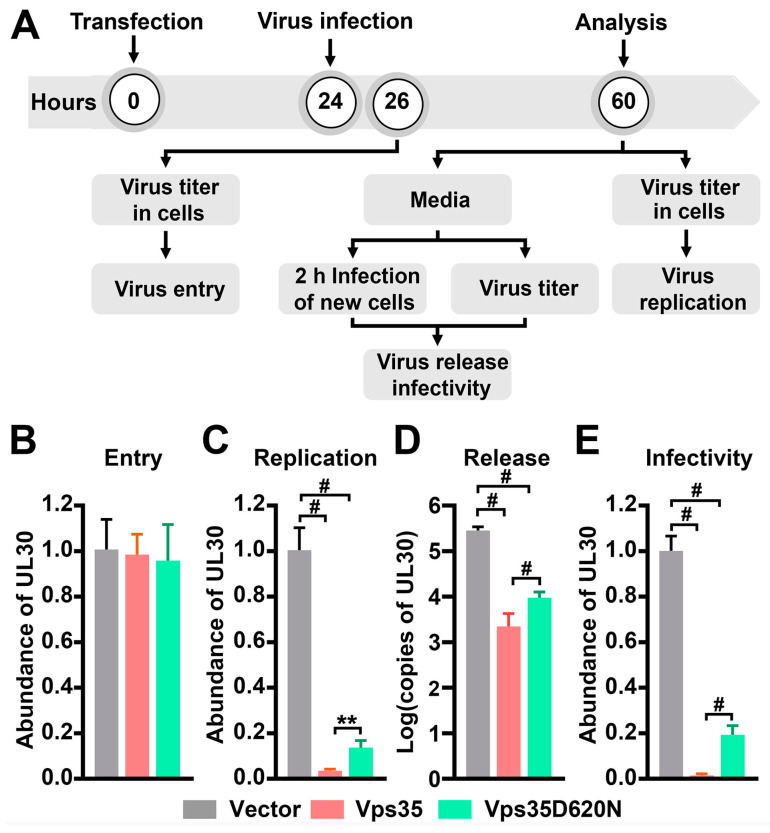
The D620N mutation in the vacuolar protein sorting 35 ortholog (Vps35) imparts a partial loss of function on Herpes simplex virus type 1 (HSV-1) propagation. (**A**) Outline of experimental schedules to detect effects of Vps35 with or without D620N on HSV-1 entry (**B**) into cells upon exposure for 2 h, replication (**C**) inside cells, and release (**D**,**E**) into culture media within 36 h after viral exposure. Relative abundance of *UL30* in genomic DNAs prepared from infected cells was determined by semi-qPCR (quantitative polymerase chain reaction) with *GAPDH* as an internal control and used as a measure of HSV-1 virion titers inside cells. Titers of virions released into culture media (conditioned media) were determined by measuring the absolute copy numbers of *UL30* in conditioned media (**D**) and relative abundance of *UL30* inside new cells exposed to conditioned media for 2 h (**E**). The latter was also utilized as a measure of **infectivity** of virions in conditioned media. Initial viral infection was conducted with a multiplicity of infection (MOI) of 3. Experiments were repeated three times. Data are Mean ± SD. One-way ANOVA followed by Turkey’s analysis for (**C**, F_(2, 20)_ = 628.4, *p* < 0.0001) and (**E**, F_(2, 24)_ = 1252, *p* < 0.0001) and by Tamhane post hoc multiple comparison for (**D**, F_(2, 25)_ = 179.3, *p* < 0.0001) were conducted to determine the statistical significance. ** *p* < 0.01; # *p* < 0.0001.

**Figure 2 cells-10-00746-f002:**
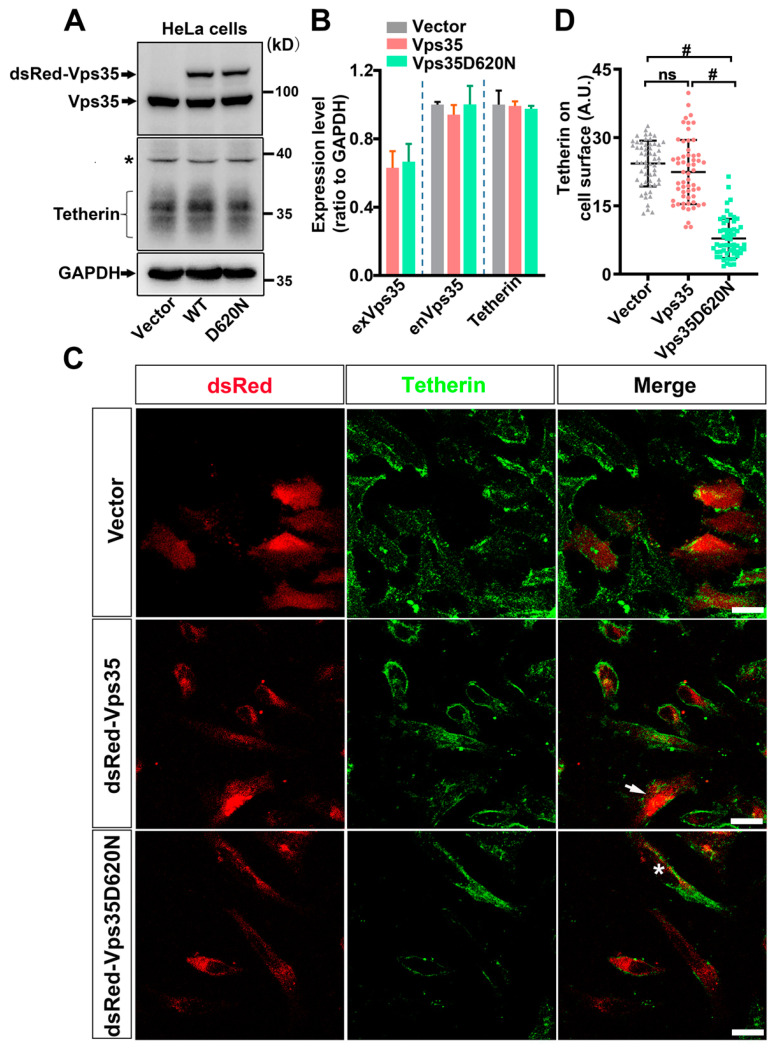
Vps35 with D620N diminishes Tetherin on cell surfaces in HeLa cells. Western blot (**A**) followed by densitometry analysis (**B**) showed that Vps35 with or without D620N was expressed at comparable levels upon transfection of HeLa cells with their expressing plasmids. * in (**A**) indicates a protein cross-reactive to the anti-Tetherin antibody. In (**B**), exVps35 represents exogenous Vps35, whereas enVps35 stands for endogenous Vps35. Neither ectopic Vps35 nor Vps35D620N affected overall expression levels of endogenous Vps35 or Tetherin. However, immunofluorescence labeling of surface-localized Tetherin (**C**) followed by densitometry analysis of imaged cells (**D**) revealed that HeLa cells transiently expressing dsRed–Vps35D620N had less Tetherin on the cell surface than cells ectopically expressing dsRed–Vps35. Cell transfections in both (**A**,**C**) were repeated three times. The star symbol beside the gel graph in (**A**) indicates a protein band cross-reactive to the antibody for Tetherin. The star symbol in (**C**) indicates that a cell expressing Vps35D620N has higher signals of Tetherin signals than neighbor cells, whereas the arrow points to a cell expressing Vps35 exhibiting lower signals of Tetherin than neighbor cells. Scale bars: 20 μm. Each symbol in the graph in (**D**) represents one cell. Data are Mean ± SD. Statistical significance in (**B**,**D**) was determined by one-way ANOVA followed by Turkey’s analysis. In (**D**), F_(2, 168)_ = 148.5, # *p* < 0.0001.

**Figure 3 cells-10-00746-f003:**
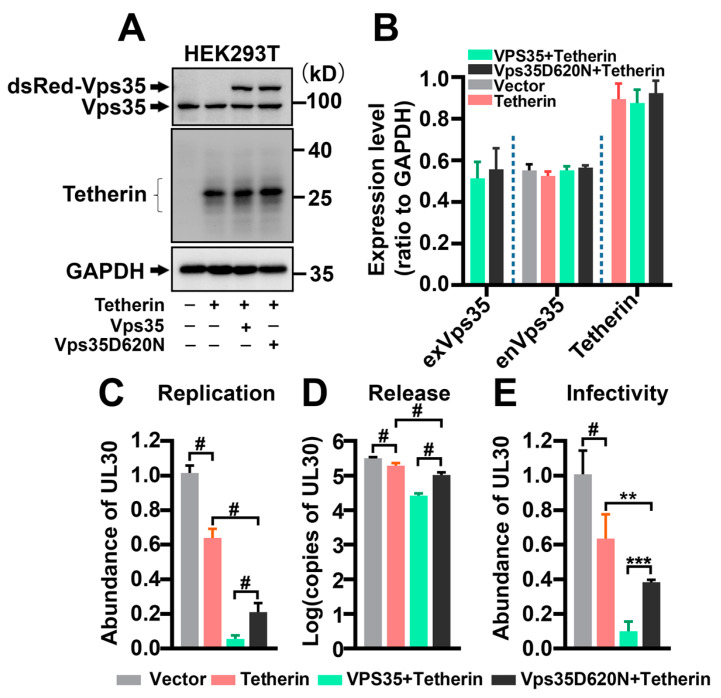
Tetherin mediates the inhibitory effect of Vps35 on HSV-1 propagation. (**A**) Western blot analysis showed that HEK293T did not express Tetherin at steady state and expressed comparable levels of endogenous Vps35 as well as exogenous Tetherin, dsRed–Vps35, and dsRed–Vps35D620N. (**B**) Densitometry quantification for (**A**) from 3 independent experiments. Ectopic expression of Tetherin in HEK293T cells suffocated HSV-1 virion replication (**C**) and release (**D** and **E**). Co-expression of Vps35 or Vps35D620N augmented the restricting activity of Tetherin on HSV-1 (**D**,**E**). Experiments were repeated three times. Data are Mean ± SD. Statistical significance in (**C**–**E**) was determined by one-way ANOVA followed by Turkey’s analysis. F_(3, 20)_ = 573.3, *p* < 0.0001 for (**C**); F_(3, 28)_ = 215.5, *p* < 0.0001 for (**D**); F_(3, 20)_ = 84.66, *p* < 0.0001 for (**E**); ** *p* < 0.01; *** *p* < 0.001; # *p* < 0.0001.

**Figure 4 cells-10-00746-f004:**
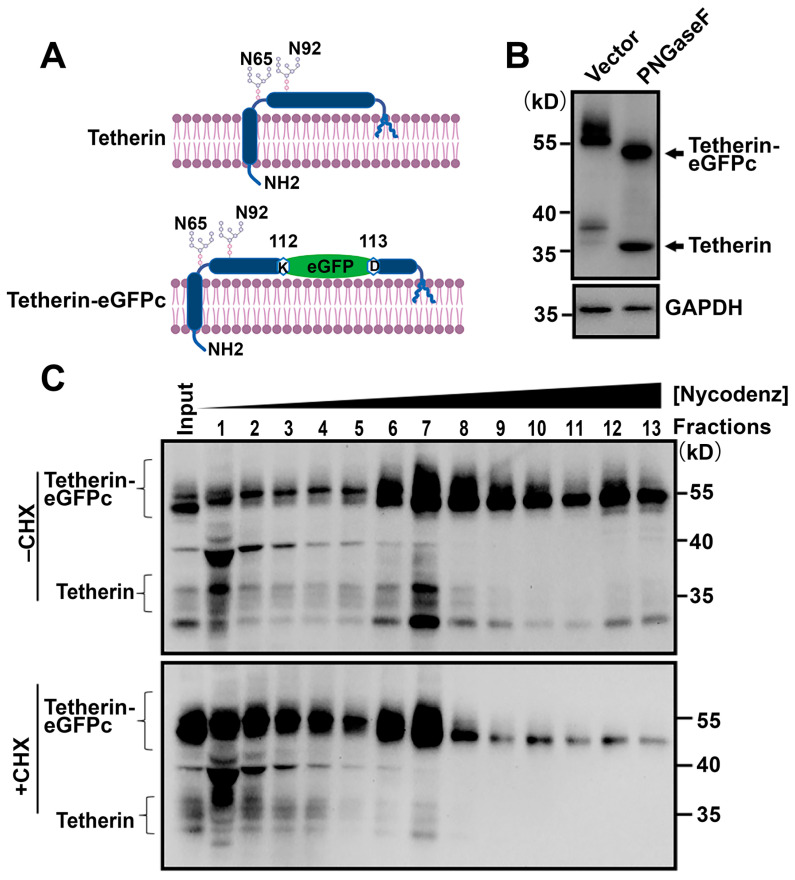
Tetherin–eGFPc mimics endogenous Tetherin. (**A**) Strategy of constructing fluorescent Tetherin reporter, Tetherin–eGFPc. An eGFP moiety is inserted between K112 and D113 of Tetherin. Also shown are the two glycosylated asparagine residues (N65 and N92). (**B**) Insertion of eGFP into Tetherin does not affect glycosylation. Total cellular membranes were prepared from HeLa cells expressing Tetherin–eGFPc and treated with or without (Mock) PNGaseF. After PNGaseF treatment, proteins were precipitated and analyzed by Western blot with indicated antibodies. GAPDH was a loading control. (**C**) Tetherin–eGFPc exhibits a distribution pattern similar to endogenous Tetherin. HeLa cells transiently expressing Tetherine–GFPc were treated with cycloheximide (CHX) to shut down de novo protein synthesis and mechanically homogenized in buffers without any detergent. After removing cell debris and nuclei, supernatants were overlaid on Nycodenz gradients, centrifuged, and fractioned. Equal volume of fractions was used for precipitating proteins and analyzed by Western blot with antibodies against Tetherin.

**Figure 5 cells-10-00746-f005:**
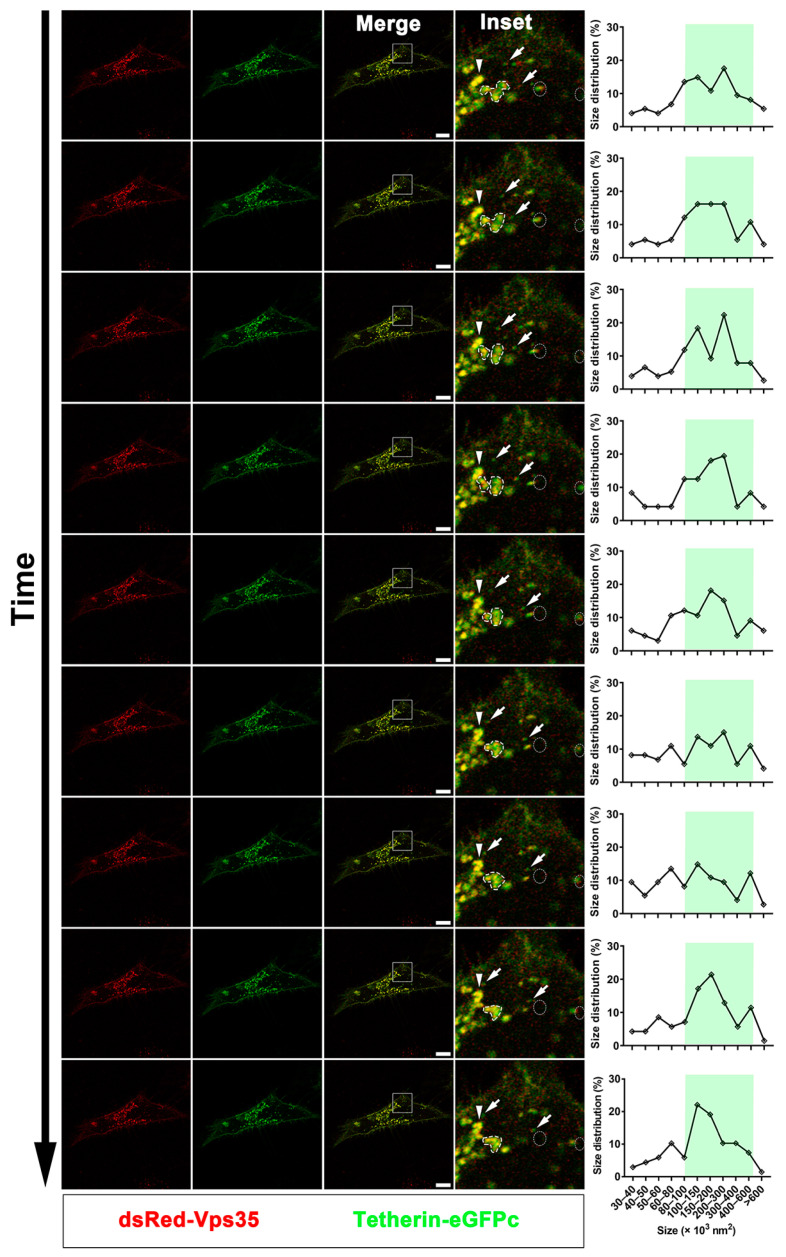
Tetherin–eGFPc traffics together with Vps35, and the D620N mutation in Vps35 perturbs the dynamics of retromer endosomes. HeLa cells were transfected with plasmids expressing Tetherin-eGFPc along with plasmids expressing dsRed–Vps35 or dsRed–Vps35D620N and imaged in real-time after being treated with cycloheximide. A series of 9 consecutive frames were chosen from movies of cells expressing dsRed-Vps35 or dsRed-Vps35D620N to highlight co-trafficking of Tetherin–eGFPc with dsRed–Vps35 or dsRed–Vps35D620N and the dynamics of Vps35 and Vps35D620N endosomes. The size of structures in each frame were measured and graphed after being manually grouped. Boxed regions in merged frames were enlarged and shown as insets. Arrows in insets trace motile structures containing both Tetherin and Vps35, whereas arrowheads indicate stationary structures. Dashed circles indicate appearances or disappearances of motile structures containing both Tetherin and Vps35, whereas dashed contours indicate morphology changes of large tubulovesicular structures where fission and fusion events were observed. Green highlights in line graphs show frequent changes in size of Tetherin-containing structures in a cell co-expressing dsRed–Vps35, whereas yellow highlights show little or no change in size of structures in a cell co-expressing dsRed–Vps35D620N. Scale bars: 10 μm.

**Figure 6 cells-10-00746-f006:**
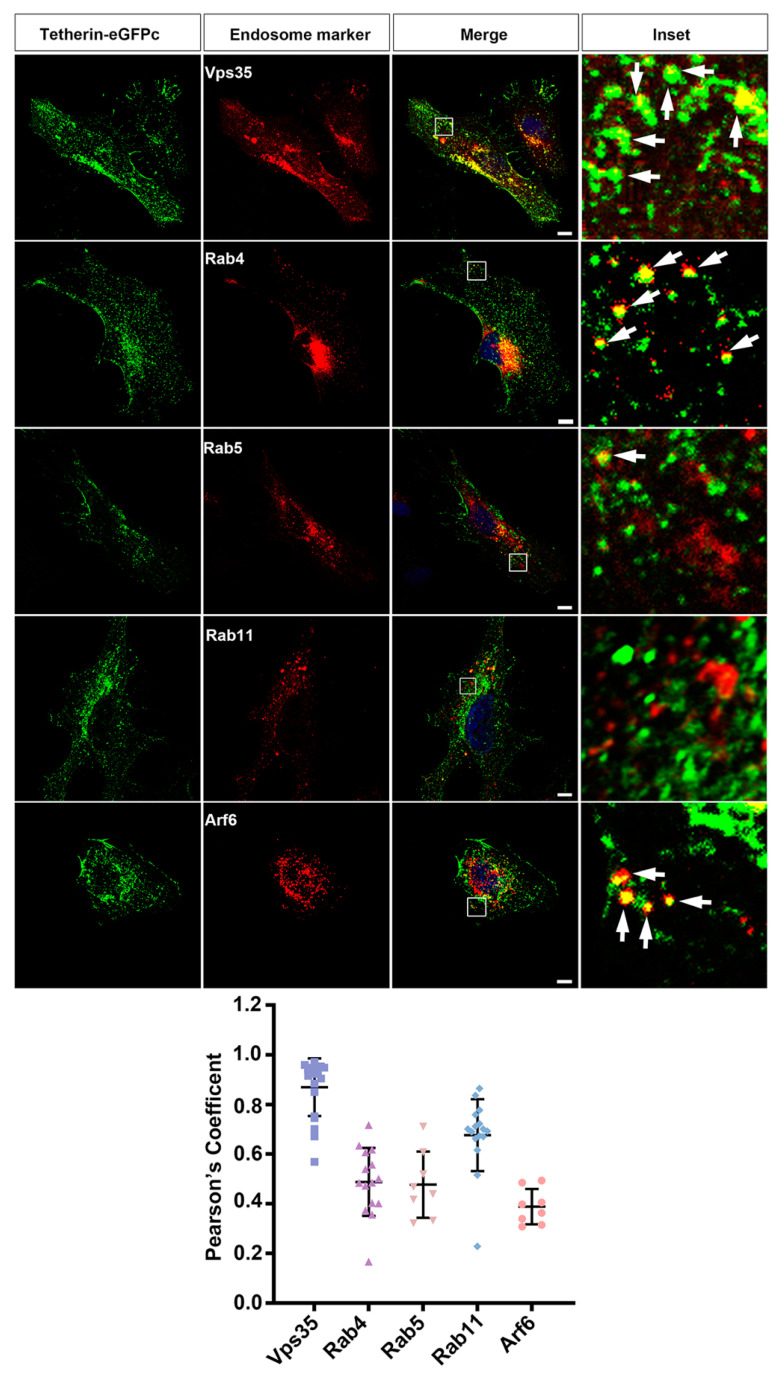
Tetherin recycles mainly via the retromer-dependent trafficking pathway. HeLa cells on glass coverslips were transfected with plasmids expressing Tetherin–eGFPc along with plasmids expressing dsRed–Vps35, dsRed–Rab4, dsRed–Rab5, dsRed–Arf6, or mCherry–Rab11. After being treated with cycloheximide, cells were fixed and processed for fluorescence microscopy. Shown are confocal images. Boxed regions in merged images were enlarged. Arrows in Insets indicate structures where Tetherin–eGFPc and a particular endosome marker protein were co-localized. Scale bars: 10 μm. Plots beneath are quantitative data of co-localization between Tetherin–GFPc and corresponding endosomal markers. Analysis was conducted with the JACoP plugin of NIH ImageJ/Fiji. Each symbol represents a cell.

## Data Availability

Not applicable.

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
