# Peer review of "Parkinson’s Disease Causative Mutation in Vps35 Disturbs Tetherin Trafficking to Cell Surfaces and Facilitates Virus Spread"

_cells, 2021, doi:10.3390/cells10040746_

Round 1
Reviewer 1 Report
This paper presents a case for involvement of Vps35 and retromer in the HSV replication and extends the study into the PD Vps35 variant, Vps36 D620N. The data is clearly presented and methodology is clearly explained. There are several issues authors need to address prior to publication:
- transfection and HSV-1 infection efficiency in repeated experiments needs to be stated as the data is based on these parameters and output can be variable if efficiency is variable
- Figure 1 shows only about 20% recovery of HSV-1 replication, release and infectivity when D620N mutant is expressed - can authors comment why the recovery is not higher? Is endogenous Vps35 levels hindering the recovery?
- In Figure 3 - tetherin is transfected in Hek293 cells but has the same molecular weight as endogenous tetherin in HeLa cells (Fig 2)?
- If overexpression of Vps35 is driving the increase in levels of Tetherin at the plasma membrane why does overexpression of Tetherin itself not have the same effect on HSV replication, release and infectivity? Figure 6 would suggest that exogenous Tetherin is mostly localised in intracellular structures?
- Figure 6 does not show any plasma membrane labelling of tetherin and Vps35 when co transfected (fixed cells) while clear PM labelling is shown in live cells - please elaborate.
- Co-localisation of Tetherin with different structures needs to be quantified and compared between Vps35 wt and mutant; Golgi labelling should also be included as overexpression of Tetherin shows clear peri-nuclear localisation
- Conclusion of Tetherin being a cargo of retromer can not be made with the experiments presented - they are co-localised on the same intracellular structures but without binding and knockdown experiments Tetherin can not be named a retromer cargo; amend the conclusions or perform the appropriate experiments
Author Response
This paper presents a case for involvement of Vps35 and retromer in the HSV replication and extends the study into the PD Vps35 variant, Vps36 D620N. The data is clearly presented and methodology is clearly explained. There are several issues authors need to address prior to publication:
transfection and HSV-1 infection efficiency in repeated experiments needs to be stated as the data is based on these parameters and output can be variable if efficiency is variable
We achieved 60% to 80% transfection efficiency in our experiments. We noted this in the Methods.
Figure 1 shows only about 20% recovery of HSV-1 replication, release and infectivity when D620N mutant is expressed - can authors comment why the recovery is not higher? Is endogenous Vps35 levels hindering the recovery?
We noticed this and therefore designed re-infection assay. Conditioned media might contain both naked viral DNAs (dead virus) and live virions, thus reducing the detection accuracy.
In Figure 3 - tetherin is transfected in Hek293 cells but has the same molecular weight as endogenous tetherin in HeLa cells (Fig 2)?
According to data in Figure 2A and Figure 3A, the molecular weight of ectopically expressed tetherin in HEK293 cells was similar to that of endogenous tetherin in HeLa cells.
If overexpression of Vps35 is driving the increase in levels of Tetherin at the plasma membrane why does overexpression of Tetherin itself not have the same effect on HSV replication, release and infectivity?
We discussed other mechanisms in the Discussion.
Figure 6 would suggest that exogenous Tetherin is mostly localised in intracellular structures? Figure 6 does not show any plasma membrane labelling of tetherin and Vps35 when co transfected (fixed cells) while clear PM labelling is shown in live cells - please elaborate.
We did not quantify the percentage of cell surface versus intracellular Tetherin. However, the images in Figure 6 did show tetherin signals at plasma membranes as clear sharp edges. We re-adjusted bright and contrast to show clearer PM labeling.
Co-localisation of Tetherin with different structures needs to be quantified and compared between Vps35 wt and mutant; Golgi labelling should also be included as overexpression of Tetherin shows clear peri-nuclear localisation
As suggested, we included quantification using NIH ImageJ JACoP in revised Figure 6B. We also compared co-localization of tetherin between Vps35 wt and mutant in supplementary figure S2 and with Golgi marker protein GM130 in supplementary figure S3.
Conclusion of Tetherin being a cargo of retromer can not be made with the experiments presented - they are co-localised on the same intracellular structures but without binding and knockdown experiments Tetherin can not be named a retromer cargo; amend the conclusions or perform the appropriate experiments
We changed the statement “a cargo of retromer” into “travels/recycles through/via the retromer labeled transport pathway”.
Reviewer 2 Report
The authors do appear to have a phenotype that is associated with Vps35D620N however many of the experiments have design issues (or lack of detail) which cast doubt on the findings. A major concern is that the ectopic expression of the wildtype fusion proteins is not neutral and the functionality of the expressed retromer fusion proteins is not characterized. A partial characterization of the tetherin fusion proteins did reveal major issues with the synthesis/folding of the protein which impacts on its cellular distribution.
The manuscript is unfocused with it attempting to make conclusions about viral infection, parkinsons and retromer dependent trafficking. The manuscript needs to be focused before publication can be considered. As a minimum for the retromer dependent trafficking the impact of KO of retromer needs to be examined or a direct interaction with the cargo of interest.
Specific Points
- Authors need to update the introduction to capture the latest D620N and retromer manuscripts.
- Please confirm that the mock transfection performed in figure one is the expression of deRed alone.
- Can you please discuss any changes in cell number/proliferation/death during the time course outlined in Figure one.
- Figure 2 – please confirm that you got an increase in “retromer” protein when the single subunit Vps35 was overexpressed. This can be done by quantifying one of the other subunits Vps26 or Vps29.
- Figure 2C – please show the single channels for each fluorophore. This will help illustrate any change in distribution. Please also include the dsRed only expressing control cells.
- Please document the control cell surface levels of tetherin (figure 2D) so we can determine which condition has changed relative to this control.
- Capture time needed for video microscopy
- Fig 5: The overlap of the red and green in the video microscopy is absolute with the majority of signal intracellular. Endosomes are dynamic (both spatially and temporally) so the should be changing position and content during these movies. In the absence of this it raises technical concerns or protein aggregates or bleed through during capture.
- Need to discuss tetherin intracellular domain and if it has any sorting signal that could bind to retromer associated cargo binding proteins.
Author Response
Reviewer-2
The authors do appear to have a phenotype that is associated with Vps35D620N however many of the experiments have design issues (or lack of detail) which cast doubt on the findings. A major concern is that the ectopic expression of the wildtype fusion proteins is not neutral and the functionality of the expressed retromer fusion proteins is not characterized. A partial characterization of the tetherin fusion proteins did reveal major issues with the synthesis/folding of the protein which impacts on its cellular distribution.
The manuscript is unfocused with it attempting to make conclusions about viral infection, parkinsons and retromer dependent trafficking. The manuscript needs to be focused before publication can be considered. As a minimum for the retromer dependent trafficking the impact of KO of retromer needs to be examined or a direct interaction with the cargo of interest.
Specific Points
Authors need to update the introduction to capture the latest D620N and retromer manuscripts.
We cited in the Discussion the latest D620N on vesicular trafficking, which was published during the preparation of this manuscript. We now noted this in the introduction as suggested.
Please confirm that the mock transfection performed in figure one is the expression of deRed alone.
We confirmed the mock transfection in Figure 1 is the expression of dsRed alone. We also noted this in revised Figure 1.
Can you please discuss any changes in cell number/proliferation/death during the time course outlined in Figure one.
As shown in Figure 1A, cells were cultured for more than 72 hours (overnight culture before transfection plus 60 hours after transfection) in an experiment. We did not observe dramatic cell death. In contrast, cells reached nearly full confluency when being collected for analyses. There was no apparent difference in cell density (confluency) or in protein concentrations between Vps35 wt and mutant transfections.
Figure 2 – please confirm that you got an increase in “retromer” protein when the single subunit Vps35 was overexpressed. This can be done by quantifying one of the other subunits Vps26 or Vps29.
The antibodies were back ordered, making us unable to conduct such studies.
Figure 2C – please show the single channels for each fluorophore. This will help illustrate any change in distribution. Please also include the dsRed only expressing control cells.
As suggested, we split the channels.
Please document the control cell surface levels of tetherin (figure 2D) so we can determine which condition has changed relative to this control.
As suggested, we included these data in revised Figure 2, C and D.
Capture time needed for video microscopy
Each cell was imaged for 5 minutes. We indicated this on the last sentence of “Immunofluorescence microscopy and time-lapse live cell imaging” in Methods.
Fig 5: The overlap of the red and green in the video microscopy is absolute with the majority of signal intracellular. Endosomes are dynamic (both spatially and temporally) so the should be changing position and content during these movies. In the absence of this it raises technical concerns or protein aggregates or bleed through during capture.
We revised the boxed areas in revised Figure 5 to show position changes of motile vesicles and morphology changes of “spatially stationary” tubulovesicular structures, which revealed fission/fusion events.
We believe that the overlap of the red and green in the video images was not a consequence of protein aggregates or bleed through, because positions of small motile vesicles changed over time, and fission and fusion events were observed, e.g., tubulovesicular structures indicated by dashed contours in the revised Figure 5.
Need to discuss tetherin intracellular domain and if it has any sorting signal that could bind to retromer associated cargo binding proteins.
As suggested, we discussed this in the Discussion “However, it is not clear whether Tetherin directly binds to Vps35 or the cargo recognition core of the retromer, as the NH2-terminal tail of Tetherin is located in the cytoplasm and does not contain a known signal for recognition by retromer [3].”. We also changed our statement “a cargo of retromer” into “travels/recycles through/via the retromer-labeled transport pathway”.
Reviewer 3 Report
Manuscript ID: cells-1109208; Yingzhuo Ding et al.
Mutations in the VPS35 gene at the PARK17 locus, encoding a component of the retromer complex, cause late-onset, autosomal dominant Parkinson’s disease (PD). However, the molecular mechanisms are not fully understood. In this paper, the authors investigated whether the PD-causative D620N mutant of VPS35 precipitated herpes simplex virus (HSV) infection. More specifically, cells (Hela, 293T) pre-infected with HSV viruses were co-transfected with either wild type or D620N dsRed-Vps35, and Tetherin-eGFPc, followed by various analyses, including a time-lapse microscopy etc. Compared with the suppressive effects of wild type Vps35 on virus replication and infectivity, those of D620N VPS35 were less clear. Furthermore, Compared with cells expressing wild type-Vps35, Tetherin trafficking to cell surfaces was less clear in cells expressing D620N-Vps35, suggesting that the mutant of Vps35 disturbs Tetherin trafficking to cell surfaces to facilitate virus spread.
Overall, I agree that the paper is clear and may contribute to the research of PD, but the paper would be improved by addition of some alpha-synuclein (aS) data/discussion. My comments are as follows.
- The loss of function of D620N VPS35 to regulate Tetherin trafficking to cell surfaces is an interesting finding. However, it is logically possible that there might be other mechanisms involved.
- PARK17 is an autosomal dominant PD. How the loss of function property of VPS35 explains the autosomal dominant nature?
- Since PARK17 is characterized by aS pathology, it is naturally predicted that D620N VPS35 might interact with aS. However, aS was never referred in the paper.
Author Response
Reviewer-3
Overall, I agree that the paper is clear and may contribute to the research of PD, but the paper would be improved by addition of some alpha-synuclein (aS) data/discussion. My comments are as follows.
The loss of function of D620N VPS35 to regulate Tetherin trafficking to cell surfaces is an interesting finding. However, it is logically possible that there might be other mechanisms involved.
We discussed other possibilities in the discussion, e.g. regulating viral proteins trafficking for virion assembly.
PARK17 is an autosomal dominant PD. How the loss of function property of VPS35 explains the autosomal dominant nature?
This is a very important question. The D620N mutation may function as a dominant negative mutant. However, there is support from genetic studies that the D620N mutation may not be fully penetrated and occurs in unaffected carriers [5, 6, 50], implicating that the pathogenicity of D620N in some cases may require concerted actions of other factor(s) or facilitator(s).
Since PARK17 is characterized by aS pathology, it is naturally predicted that D620N VPS35 might interact with aS. However, aS was never referred in the paper.
As suggested, we included the connections between Vps35 and alpha-synuclein in the second paragraph of the Introduction.
Reviewer 4 Report
The authors investigated the VPS35 mutation-associated viral infection. There are some points required to be clarified and modified.
- In general, data was suggested to presented as mean+-S.D. but not SEM
- Transient over-expression may alter a lot of cellular physiological status, whereas stable over-expressing lines may be preferred.
- Work on non-neuronal cells limits the application on Parkinson's disease.
- In line 323, authors mentioned "Taken together, these data suggest that the D620N mutation in Vps35 causes loss-of-function on HSV-1 propagation". The description of "loss-of-function" is not precisely and introduce confuse.
- Figure 2C should be presented like Figure 5/6, split the red/green color image with merged one (take away blue to clearly demonstrate co-localization).
- Figure 3B: comparison between two cell types about the viral infection can not lead to the causal relationship between VPS35 with viral infection, since there are numerous differences between the tw cell types.
- Since HEK293T cells express endogenous VPS35 but not Tetherin, it would be nice to silencing/inhibiting VPS35 to investigate the effect of Tetherin alone, and the possible synergistic effect.
- Figure 5: there should be a quantification figure for the change of green area and yellow area for better readability.
- Figure 6: since the paper investigated the role of VPS35 D620N, there should be some investigations about the effect of VPS35 D620N on the Tetherin recycle pathway.
- The authors should have more discussion on the role of viral infection in PD.
Author Response
Reviewer4
The authors investigated the VPS35 mutation-associated viral infection. There are some points required to be clarified and modified.
In general, data was suggested to presented as mean+-S.D. but not SEM
As suggested, we changed SEM into SD in all graphs/plots.
Transient over-expression may alter a lot of cellular physiological status, whereas stable over-expressing lines may be preferred.
We discussed this limitation in the Discussion.
Work on non-neuronal cells limits the application on Parkinson's disease.
We discussed this in the Discussion.
In line 323, authors mentioned "Taken together, these data suggest that the D620N mutation in Vps35 causes loss-of-function on HSV-1 propagation". The description of "loss-of-function" is not precisely and introduce confuse.
As suggested, we changed the statement into “these data suggest that the D620N mutation interferes with Vps35 in dampening HSV-1 propagation”.
Figure 2C should be presented like Figure 5/6, split the red/green color image with merged one (take away blue to clearly demonstrate co-localization).
As suggested, we made these changes.
Figure 3B: comparison between two cell types about the viral infection can not lead to the causal relationship between VPS35 with viral infection, since there are numerous differences between the two cell types.
We placed the comparison data in bar graph in original Figure 3B as supplementary data in supplementary figure S1. Please be aware that revised Figure 3B were quantitative data of Western blot analysis in Figure 3A.
Since HEK293T cells express endogenous VPS35 but not Tetherin, it would be nice to silencing/inhibiting VPS35 to investigate the effect of Tetherin alone, and the possible synergistic effect.
We did not perform knockdown experiments because our data in Figure 3 C-E suggest that Vps35 and tetherin synergize in restricting virus propagation.
Figure 5: there should be a quantification figure for the change of green area and yellow area for better readability.
Since dsRed-Vps35 and tetherin-GFPc were well co-localized, no difference was found when background signals were removed. However, we revised the boxed areas to show position changes of motile vesicles and morphology changes of “spatially stationary” tubulovesicular structures, which revealed fission/fusion events.
Figure 6: since the paper investigated the role of VPS35 D620N, there should be some investigations about the effect of VPS35 D620N on the Tetherin recycle pathway.
We showed in live cell imaging in Figure 5 that Vps35 D620N disturbed endosomal dynamics.
We also included new data to compare the co-localization of tetherin-GFPc between Vps35 wt and mutant in Figure S2. The data showed that signals of tetherin-GFPc and dsRed-Vps35D620N were concentrated at large structures when compared with relatively diffuse structures containing dsRed-Vps35 and tetherin-GFPc.
However, it is not clear to us whether Tetherin-GFPc was shifted to other endosomal compartments away from the wildtype retromer dependent transport pathway.
The authors should have more discussion on the role of viral infection in PD.
As suggested, we included likely mechanisms of viral infection in PD pathogenesis.
Round 2
Reviewer 1 Report
/Please note that authors did address most of the concerns but have unfortunately not addressed some of the larger ones:/
/Figure 1 shows only about 20% recovery of HSV-1 replication, release and infectivity when D620N mutant is expressed - can authors comment why the recovery is not higher? Is endogenous Vps35 levels hindering the recovery? /
We noticed this and therefore designed re-infection assay. Conditioned media might contain both naked viral DNAs (dead virus) and live virions, thus reducing the detection accuracy. - this response makes not links to the endogenous levels of Vps35 or now this discrepancy may be addressed outside of the re-infection assay.
/In Figure 3 - tetherin is transfected in Hek293 cells but has the same molecular weight as endogenous tetherin in HeLa cells (Fig 2)?/
According to data in Figure 2A and Figure 3A, the molecular weight of ectopically expressed tetherin in HEK293 cells was similar to that of endogenous tetherin in HeLa cells.- the MW of Tetherin seems to be the same when fused to eGFP (in Hek293 cells) and when endogenous is labelled in HeLa cells - I was suggesting that maybe the MW labelling needs to be checked in Fig 3A
/If overexpression of Vps35 is driving the increase in levels of Tetherin at the plasma membrane why does overexpression of Tetherin itself not have the same effect on HSV replication, release and infectivity? /
We discussed other mechanisms in the Discussion. - this was actually not discussed - can the author point out the further discussion they are referring to?
Also, the following phrase needs to change to:
We changed the statement “a cargo of retromer” into “travels/recycles through/via the retromer labeled transport pathway” “trafficking is mediated in retromer-dependent manner/pathway"
Author Response
/Figure 1 shows only about 20% recovery of HSV-1 replication, release and infectivity when D620N mutant is expressed - can authors comment why the recovery is not higher? Is endogenous Vps35 levels hindering the recovery? /
We apologize for misunderstanding this comment in our previous response.
Yes, endogenous Vps35 definitely played a role. Vps35D620N also possessed some degrees of normal function, though it may interfere with endogenous Vps35 as a dominant negative mutant.
/In Figure 3 - tetherin is transfected in Hek293 cells but has the same molecular weight as endogenous tetherin in HeLa cells (Fig 2)?/….- I was suggesting that maybe the MW labelling needs to be checked in Fig 3A
We were sorry for not making it clear. The construct we used for transfecting 293T cells expressed wild-type Tetherin without a GFP tag.
/If overexpression of Vps35 is driving the increase in levels of Tetherin at the plasma membrane why does overexpression of Tetherin itself not have the same effect on HSV replication, release and infectivity? / We discussed other mechanisms in the Discussion. - this was actually not discussed - can the author point out the further discussion they are referring to?
We discussed other possibilities as “Vps35 has been shown to regulate the replication, assembly, and release of virions [39-42]. The inhibitory effect of Vps35 on HSV-1 propagation might also arise from its regulation on the trafficking of HSV-1 envelop proteins required for the assembly and release of HSV-1 virions and or on the trafficking of other host virus-restricting factor(s). Future studies are needed to explore these possibilities.” in the fourth paragraph of the Discussion.
We have now made this discussion clearer by adding a sentence “However, overexpression of Tetherin alone in HEK293T cells did not produce the same effect on HSV-1 propagation as co-expression with Vps35 or with Vps35D620N, suggesting that the inhibitory effect of Vps35 on virus propagation involves other factor(s).”. This sentence was highlighted in green in the fourth paragraph of the revised Discussion.
Also, the following phrase needs to change to: We changed the statement “a cargo of retromer” into “travels/recycles through/via the retromer labeled transport pathway” “trafficking is mediated in retromer-dependent manner/pathway"
Many thanks for this suggestion.
Reviewer 2 Report
The authors have not addressed the major concerns raised in the original review of the manuscript. Without addressing these major issues my recommendations will remain unchanged.
I have not bothered to evaluate the responses to the specific points as this will not impact on the indicated major faults with this manuscript
Author Response
The authors do appear to have a phenotype that is associated with Vps35D620N however many of the experiments have design issues (or lack of detail) which cast doubt on the findings. A major concern is that the ectopic expression of the wildtype fusion proteins is not neutral and the functionality of the expressed retromer fusion proteins is not characterized. A partial characterization of the tetherin fusion proteins did reveal major issues with the synthesis/folding of the protein which impacts on its cellular distribution.
We agree with this reviewer on the limitations using the ectopic expression in functional studies of proteins, as we discussed this in the Discussion of our manuscript.
We have conducted co-immunoprecipitation to determine the functionality of ectopically expressed dsRed-Vps35. The results are now included in revised figure S1A and showed that ectopically expressed dsRed-Vps35 was co-precipitated with endogenous Vps26. The new data indicate the functionality of ectopically expressed dsRed-Vps35 in the retromer complex.
The manuscript is unfocused with it attempting to make conclusions about viral infection, parkinsons and retromer dependent trafficking. The manuscript needs to be focused before publication can be considered. As a minimum for the retromer dependent trafficking the impact of KO of retromer needs to be examined or a direct interaction with the cargo of interest.
We do not agree with this reviewer on the comment “the manuscript is unfocused with it attempting to make conclusions about viral infection, parkinsons and retromer dependent trafficking”. We investigated effects of a Parkinson-relevant mutation in Vps35 on viral infection, which is relevant to the pathogenesis of Parkinson’s disease, and trafficking of Tetherin, a virus-restricting factor implicated in previous studies to undergo recycling through the retromer dependent pathway (we cited these studies in the Introduction).
KO of retromer is an important strategy for studying the retromer dependent trafficking, though it is not the only one. We have not been successful in establishing a Vps35-KO cell line for replicating the findings in the manuscript.
As only the N-terminal 20aa tail of Tetherin is located in the cytoplasm, we analyzed human Tetherin that was used in our study and did not find the consensus recycling signal [F/Y/W]X[L/V/M] for recognition by retromer. We did not examine whether Tetherin directly interacted with retromer. In observance of lacking this knowledge, we changed our statement “Tetherin is a cargo of retromer” into “Tetherin travels/recycles through/via the retromer labeled/positive transport pathway”.
Specific Points
Authors need to update the introduction to capture the latest D620N and retromer manuscripts.
We cited in the Discussion the latest D620N on vesicular trafficking, which was published during the preparation of this manuscript. We have now noted this in the introduction as well.
Please confirm that the mock transfection performed in figure one is the expression of deRed alone.
We confirmed the mock transfection in Figure 1 is the expression of dsRed alone. We also noted this in revised Figure 1.
Can you please discuss any changes in cell number/proliferation/death during the time course outlined in Figure one.
As shown in Figure 1A, cells were cultured for more than 72 hours (overnight culture before transfection plus 60 hours after transfection) in an experiment. We did not observe dramatic cell death. In contrast, cells reached nearly full confluency when being collected for analyses. There was no apparent difference in cell density (confluency) or in protein concentrations between Vps35 wt and mutant transfections.
Figure 2 – please confirm that you got an increase in “retromer” protein when the single subunit Vps35 was overexpressed. This can be done by quantifying one of the other subunits Vps26 or Vps29.
We conducted the suggested experiments. The data were included in the right figure herein (the figure was not successfully pasted, please refer to the PDF file) and showed that transient expression of dsRed-Vps35 did not apparently change the expression level of endogenous Vps26, as proposed.
As suggested, we also conducted co-immunoprecipitation. The results were included in revised figure S1A and showed that ectopically expressed dsRed-Vps35 interacted with endogenous Vps26.
Figure 2C – please show the single channels for each fluorophore. This will help illustrate any change in distribution. Please also include the dsRed only expressing control cells.
As suggested, we split the channels.
Please document the control cell surface levels of tetherin (figure 2D) so we can determine which condition has changed relative to this control.
As suggested, we included these data in revised Figure 2, C and D.
Capture time needed for video microscopy
Each cell was imaged for 5 minutes. We indicated this on the last sentence of “Immunofluorescence microscopy and time-lapse live cell imaging” in Methods.
Fig 5: The overlap of the red and green in the video microscopy is absolute with the majority of signal intracellular. Endosomes are dynamic (both spatially and temporally) so the should be changing position and content during these movies. In the absence of this it raises technical concerns or protein aggregates or bleed through during capture.
We revised the boxed areas in revised Figure 5 to show position changes of motile vesicles and morphology changes of “spatially stationary” tubulovesicular structures, which revealed fission/fusion events.
We believe that the overlap of the red and green in the video images was not a consequence of protein aggregates or bleed through, because positions of small motile vesicles changed over time, and fission and fusion events were observed, e.g., tubulovesicular structures indicated by dashed contours in the revised Figure 5.
Need to discuss tetherin intracellular domain and if it has any sorting signal that could bind to retromer associated cargo binding proteins.
As suggested, we discussed this in the Discussion “However, it is not clear whether Tetherin directly binds to Vps35 or the cargo recognition core of the retromer, as the NH2-terminal tail of Tetherin is located in the cytoplasm and does not contain a known signal, e.g., [F/Y/W]X[L/M/V], for recognition by retromer [3].”. We also changed our statement “a cargo of retromer” into “travels/recycles through/via the retromer-labeled transport pathway”.

Reviewer 3 Report
As I commented before, the authors should refer to alpha-synuclein, even in Discussion. Otherwise, it is unclear why the mutation in VPS is linked to autosomal dominant Parkinson’s disease.
Author Response
As I commented before, the authors should refer to alpha-synuclein, even in Discussion. Otherwise, it is unclear why the mutation in VPS is linked to autosomal dominant Parkinson’s disease.
We included the connections between Vps35 and alpha-synuclein in the second paragraph of the Introduction in our first-round revision, which were highlighted in yellow.
We have now added a new paragraph in the Discussion to extend the likely interplay among Vps35 D620N, viral infection, and alpha-synuclein. This new paragraph was highlighted in green.
Reviewer 4 Report
I endorse the revisions made by the authors
Author Response
We appreciated this reviewer for the time spent in our manuscript.